# MicroRNA-181a Regulates the Proliferation and Differentiation of Hu Sheep Skeletal Muscle Satellite Cells and Targets the YAP1 Gene

**DOI:** 10.3390/genes13030520

**Published:** 2022-03-16

**Authors:** Mingliang He, Weibo Zhang, Shan Wang, Ling Ge, Xiukai Cao, Shanhe Wang, Zehu Yuan, Xiaoyang Lv, Tesfaye Getachew, Joram M. Mwacharo, Aynalem Haile, Wei Sun

**Affiliations:** 1College of Animal Science and Technology, Yangzhou University, Yangzhou 225009, China; 13013706620@163.com (M.H.); zhangwb7053@163.com (W.Z.); sunnyshan5233@163.com (S.W.); gl1024winnie@163.com (L.G.); shanhe12315@163.com (S.W.); 2Joint International Research Laboratory of Agriculture and Agri-Product Safety of Ministry of Education of China, Yangzhou University, Yangzhou 225009, China; cxkai0909@163.com (X.C.); yuanzehu1988@163.com (Z.Y.); dx120170085@yzu.edu.cn (X.L.); 3International Centre for Agricultural Research in the Dry Areas, Addis Ababa 999047, Ethiopia; t.getachew@cgiar.org (T.G.); j.mwacharo@cgiar.org (J.M.M.); a.haile@cgiar.org (A.H.)

**Keywords:** miR-181a, YAP1, proliferation, differentiation, satellite cell, Hu sheep

## Abstract

MicroRNA (miRNA) is of great importance to muscle growth and development, including the regulation of the proliferation and differentiation of skeletal muscle satellite cells (SMSCs). In our research group’s previous study, we found that miR-181a is differentially expressed in the longissimus dorsi muscle of Hu sheep at different stages. We speculated that miR-181a may participate in the growth and development process of Hu sheep. To understand the mechanism of miR-181a regulating the growth and development of Hu sheep skeletal muscle, we extracted skeletal muscle satellite cells from the longissimus dorsi muscle of 3-month-old Hu sheep fetuses and performed a series of experiments. Our results showed that miR-181a suppressed SMSCs’ proliferation using QRT-PCR, Western blot, CCK-8, EDU, and Flow cytometry cycle tests. In addition, QRT-PCR, Western blot, and immunofluorescence indicated that miR-181a facilitated the differentiation of SMSCs. Then, we used dual-luciferase reporter gene detection, QRT-PCR, and Western blot to find that the Yes1-related transcription regulator (YAP1) is the target gene of miR-181a. Our study supplies a research basis for understanding the regulation mechanism of miR-181a on the growth of Hu sheep skeletal muscle.

## 1. Introduction

As a native sheep bred in China, the Hu sheep (Ovis aries) has many excellent traits, such as good reproductive performance, high meat–bone ratio, and tender, juicy meat. As we know, the growth of muscle determines the yield of mutton [1]. Skeletal muscle growth includes muscle cells’ formation and proliferation, as well as myotubes and muscle fibers’ formation [2,3]. Here, we explored the regulation of miR-181a on the skeletal muscle satellite cells of Hu sheep to understand its effect on skeletal muscle. The previous RNA-seq of the skeletal muscle of sheep before and after birth also showed differential expression of miR-181a [4]. Skeletal muscle satellite cells are a kind of static, mononuclear myogenic cell [5]. They participate in skeletal muscle regeneration and growth.

MicroRNAs (miRNAs) are a kind of non-coding, single-stranded, small RNA sequence, which can be involved in post-transcriptional regulation, and which do not have the ability to encode proteins. As we know, miRNAs can target mRNA and degrade or suppress it, reducing mRNA level and translation efficiency [6]. With the increasing research on miRNAs, they have been found to be widely involved in various cellular processes, and to play important roles, including cell proliferation, cell differentiation, and cell apoptosis [7,8]. MiRNA can also regulate the growth and development of ruminant species’ skeletal muscle. MiR-143 can target IGFBP5 and suppress the proliferation and differentiation of bovine SMSCs [9]. MiR-27b can suppress the proliferation of SMSCs and facilitate SMSCs’ differentiation by targeting Pax3 [10]. These studies have shown that miRNA is very important in ruminant skeletal muscle. However, the study of miRNA on the muscle of Hu sheep needs further exploration.

MiR-181a is a member of the miR-181 family, and the family is conserved among different species [11]. The miR-181a family consists of four mature members: miR-181a, miR-181b, miR-181c, and miR-181d [12]. MiR-181a has been reported to participate in many life activities. In cancer progression, miR-181a has been found to target GAS7 and to facilitate gefitinib resistance in non-small-cell lung cancer cells [13]. In skeletal muscle studies, up-regulation of miR-181a can improve mitochondrial mass and muscle function in aged mice [14]. In ruminants, miR-181a has been reported to regulate lipid synthesis negatively in bovine mammary cells by targeting ACSL1 [15]. However, there are no research reports on miR-181a in the skeletal muscle of Hu sheep.

Generally, miRNAs play a regulatory role by binding to the 3′UTR (untranslated region) of mRNA [16]. So, we predicted the target gene of miR-181a using RNAhybrid online software for understanding the regulatory mechanism of miR-181a on downstream genes. The result of software prediction showed that miR-181a could target the YAP1 gene. As a transcriptional co-activator, YAP1 is mainly regulated by the Hippo pathway [17]. Generally, YAP1 binds to the target DNA region by cooperating with transcription factors, and further activates downstream transcription to transduce proliferation signals [18]. In skeletal muscle, YAP1 also plays an important role. YAP1 has been found to facilitate the proliferation of myoblasts and satellite cells [19,20]. In our study, to further understand the functions of miR-181a on the SMSCs of Hu sheep, we explored whether miR-181a could target YAP1.

The purpose of our research was to understand the function of miR-181a on the proliferation and differentiation of SMSCs in Hu sheep. This study will contribute to the understanding of the function of miRNAs in Hu sheep skeletal muscle, and provides a theoretical basis for Hu sheep molecular breeding.

## 2. Methods

### 2.1. Ethics Statement

All experimental schemes were implemented in accordance with the “Jiangsu Province laboratory animal management measures”, and approved by the Animal Ethics Committee of Yangzhou University (Approval number: No. 202103279).

### 2.2. Experimental Animals and Tissues

Sheep for tissue collection were provided by Suzhou Sheep Farm (Suzhou, Jiangsu, China). After the ewe was anesthetized with 2% lidocaine hydrochloride (240 mg, 4 mg/kg), a 3-month-old Hu sheep fetus was collected by surgery; 2 g of skeletal muscle was then taken from the longissimus dorsi muscle.

### 2.3. Cell Isolation and Culture

The complete medium used for HEK293T cells includes DMEM High-Glucose medium (Sigma-Aldrich, St. Louis, MO, USA), 10% fetal bovine serum (Gibco, Grand Island, NY, USA), and 1% penicillin-streptomycin-amphotericin (Solarbio, Beijing, China).

The SMSCs are separated according to the method previously described [21]. When culturing SMSCs in different stages, we use a different medium. The growth medium includes DMEM-F12 (Sigma-Aldrich, St. Louis, MO, USA), 10% fetal bovine serum, and 1% penicillin–streptomycin–amphotericin; the differentiation medium includes DMEM-F12, 2% horse serum (Solarbio, Beijing, China), and 1% penicillin–streptomycin–amphotericin.

All cell culture conditions are 37 °C, 5% CO_2_.

### 2.4. Total RNA of SMSCs Extraction, cDNA Synthesis and qRT-PCR

Total RNA of SMSCs was extracted by Trizol (Takara, Dalian, China) and stored in liquid nitrogen. We used the miRNA 1st strand cDNA Synthesis Kit (by stem-loop) (Vazyme, Nanjing, China) to synthesize miA cDNA, and we used miRNA Universal SYBR qPCR Master Mix (Vazyme, Nanjing, China) to detect the expression of microRNA-181a in cells. We used a one-step reverse transcription Kit (Tiagen, Beijing, China) to synthesize gene cDNA, and the 2 × TSINGKE^®^ Master qPCR Mix (Tsingke, Nanjing, China) was used to detect the expression levels of related genes in cells. A real-time fluorescence quantitative instrument (Bio-Rad, Hercules, CA, USA) was used to perform QRT-PCR. Sheep GAPDH and U6 were used as reference genes. Each sample was performed in three repeated tests. The 2^−ΔΔ CT^ method was used to calculate relative expression [22].

### 2.5. Primers for qRT-PCR

Primers of miR-181a and U6 were designed using miRNA Design V1.01 software (Vazyme, Nanjing, China). U6 was used as a housekeeping gene for miRNA expression detection. Premier Primer 5.0 software (Premier Biosoft International, Palo Alto, CA, USA) was used to design primers of related genes. GAPDH was used as a housekeeping gene for gene expression detection. These primers were synthesized by TsingKe (Nanjing, China). Primers are shown in Table 1 and Table 2.

### 2.6. Oligonucleotides and Construction of Related Plasmids

MiR-181a mimic, mimic-NC (negative control), miR-181a inhibitor and inhibitor NC were designed and synthesized by RiboBio (Guangzhou, China). Oligonucleotide sequences are shown in Table 3.

We used PCR to amplify the 3′UTR region of the YAP1 gene, which contains the predicted binding site of miR-181a and YAP1. Then, we used the homologous recombination method to clone the amplified fragment into a PMIR dual-luciferase reporter vector. The restriction sites are HindIII and mIuI. Finally, we confirmed the successful insertion of the target fragment by sequencing the constructed plasmid. In order to construct a mutant-type dual-luciferase reporter plasmid, we used the Fast Mutagenesis Kit V2 (Vazyme, Nanjing, China) according to the instructions. Then, we sequenced the constructed mutant plasmid, and the result showed TTGGATG mutated to ACCTGCA. Primers are shown in Table 4.

### 2.7. Cell Transfection

The reagent used for cell transfection in this experiment is jetPRIME transfection reagent (Polyplus, Illkirch, France). In accordance with the instructions, we confirmed the transfection operations and reagent dosage.

### 2.8. Dual-Luciferase Reporter Assay

In the double luciferase experiment, HEK293T cells were used as tool cells. A 24-well cell culture dish was used to culture HEK293T cells. After the cells’ density reached 60%, we mixed the wild-type vector (PMIR-YAP1-3′UTR-WT) or the mutant vector (PMIR-YAP1-3′UTR-MT) with miR-181a mimic or NC, respectively. We then transfected the mixture into HEK293T cells. After 48 h of transfection, we used a dual-luciferase detection kit (Vazyme, Nanjing, China) to process the HEK293T cells. After the cells were processed, the luciferase activity was detected by a multi-mode micropore detection system (EnSpire, PerkinElmer, Waltham, MA, USA).

### 2.9. CCK-8 Assay

A 96-well cell culture dish was used to culture the SMSCs. After the cells’ density reached 40%, we transfected miR-181a mimic and mimic NC into the SMSCs. We used the CCK-8 Kit (Vazyme, Nanjing, China) to detect the absorbance of cells at 450 nm at 0 h (12 h after transfection), 24 h, 48 h, and 72 h. The absorbance-detection instrument is the multimode micropore detection system (EnSpire, PerkinElmer, USA).

### 2.10. EdU Assay

A 24-well cell culture dish was used to culture the SMSCs. After the cells’ density reached 40%, we transfected miR-181a mimic and mimic NC into the SMSCs. After 48 h of transfection, we incubated the cells with EDU for two hours. Then, we used 1X PBS (Solarbio, Beijing, China) to wash the cells. Next, we used 4% paraformaldehyde (Solarbio, Beijing, China) to fix SMSCs for 0.5 h. Finally, we used EdU Apollo In Vitro Imaging Kit (RiboBio, Guangzhou, China) to treat the SMSCs. A fluorescence inverted microscope (Nikon, Tokyo, Japan) was used for cell image acquisition. Image-Pro software was used for data analysis. In each group of treatments, three areas were randomly selected to calculate the number of stained cells.

### 2.11. Cell Cycle Assay

A 6-well cell culture dish was used to culture SMSCs. After the cells’ density reached 40%, we transfected miR-181a mimic and mimic NC into the SMSCs. After 2 days of transfection, we collected the cells, added 70% ethanol, and fixed them overnight at 4 °C. Then, we used Cell Cycle Kit (Beyotime, Shanghai, China) to treat the SMSCs. Finally, an FACSAria SORP flow cytometer (BD company, Franklin, NJ, USA) was used to analyze cells. ModFit LT software was used for data analysis.

### 2.12. Immunofluorescence Assay

A 12-well cell culture dish was used to culture SMSCs. The cell culture medium was the growth and differentiation medium. We transfected miR-181a mimic and mimic NC into the SMSCs after the cells reached 40% density. After 1 day of transfection, we replaced the growth medium with a differentiation medium. After 3 days of differentiation, we aspirated the medium and washed the SMSCs with PBS (Solarbio, Beijing, China). Then, we fixed the SMSCs with 4% paraformaldehyde (Solarbio, Beijing, China) for 30 min. Next, we used 0.5% Triton X-100 (Solarbio, Beijing, China) to permeate the SMSCs for 15 min. After 15 min of infiltrating the SMSCs, we used 1% BSA (Solarbio, Beijing, China) to incubate the SMSCs for 60 min at 37 °C. Finally, we used the primary antibody (MyHC, Affinity Biosciences, Cincinnati, OH, USA, 1:400) to incubate the SMSCs in darkness at 4 °C overnight. We then aspirated the primary antibody and washed the SMSCs with 1X PBST (Solarbio, Beijing, China). Then, we used the secondary antibody (Goat Anti-Rabbit IgG H&L (Alexa Fluor^®^ 594), Abcam, Cambridge, UK, 1:400) to incubate the SMSCs at 37 °C for 1 h. We then aspirated the primary antibody and washed the SMSCs with 1X PBST after the secondary antibody was incubated. DAPI (Beyotime, Shanghai, China) was used to stain the nuclei of the SMSCs. Finally, a fluorescence inverted microscope (Nikon, Tokyo, Japan) was used to observe and photograph the SMSCs.

### 2.13. Western Blot

RIPA Cell Lysate (Beyotime, Shanghai, China) was used to lyse SMSCs. Then, we centrifuged the lysate cells at 12,000 rpm for 10 min and collected the supernatant. BCA protein quantification kit (Beyotime, Shanghai, China) was used to determine protein concentration. Next, we subjected the protein sample to polyacrylamide gel electrophoresis, and then transferred it to PVDF (Solarbio, Beijing, China). We used 5% skimmed milk powder to seal the PVDF membrane, which was closed at 25 °C for 1.5–2 h. We then added the primary antibody and incubated it overnight at 4 °C. The next day, we used 1X TBST (Solarbio, Beijing, China) to wash the PVDF membrane. After wash-over, we incubated cells with the secondary antibody at 25 °C for 1 h. We used Electrochemiluminescence (ECL) for imprinting display. The ChemDoc^TM^Touch Imaging System (Bio-Rad, Hercules, CA, USA) was used for relative expression of the protein obtained. The antibodies used in this study and their dilution ratios are as follows: Anti-CDK2 antibody (Abcam, Cambridge, UK, 1:1500), Anti-MYHC antibody (Affinity Biosciences, Cincinnati, OH, USA, 1:1000), Anti-GAPDH antibody (Abcam, Cambridge, UK, 1:1000), Goat Anti-Rabbit IgG H&L(HRP) (Abcam, Cambridge, UK, 1:5000), Rabbit Anti-Mouse IgG H&L(HRP) (Abcam, Cambridge, UK, 1:5000).

### 2.14. Statistical Analysis

Statistical analysis was performed by SPSS25.0 software (SPSS Inc., Chicago, IL, USA). The comparative analysis of the two groups was performed by an unpaired Student’s *t*-test. We considered the data to be statistically significant only when *p* < 0.05. For each group of treatments, we performed 3 repetitions, and all data are presented as least-squares means ± SEM (standard error of the mean).

## 3. Results

### 3.1. MiR-181a Suppresses SMSCs Proliferation

To reveal the regulating effect of miR-181a on the proliferation of Hu sheep SMSCs, we transfected SMSCs with miR-181a mimic, mimic NC, miR-181a inhibitor, and inhibitor NC. Results showed that overexpression of miR-181a in SMSCs facilitates its expression, while interference results in the opposite effect (Figure 1a,b). This indicates that miR-181a, mimic, miR-181a inhibitor, and NC have an effect. Then, qRT-PCR, CCK-8, EDU, flow cell-cycle detection, and Western blot were used to test the regulation of overexpression and interference of miR-181a on the proliferation of SMSCs. The experimental results indicate that overexpression of miR-181a in SMSCs can significantly suppress the expression of PCNA and CDK2 genes and the protein expression of CDK2 genes (Figure 1c,i,k), and that interference with miR-181a can significantly facilitate the mRNA expression of PCNA and CDK2 genes and the protein expression of CDK2 genes (Figure 1d,j,l), preliminarily indicating that miR-181a participates in the regulation of the proliferation of SMSCs. At the same time, both CCK-8 and EDU assays indicated that overexpression of miR-181a can suppress the proliferation of SMSCs (Figure 1e,m,o), and interference with miR-181a can facilitate the proliferation of SMSCs (Figure 1f,n,p). In addition, flow cytometry showed that overexpression of miR-181a can suppress the cell-cycle progression (Figure 1g), and interference with miR-181a can facilitate the cell-cycle progression (Figure 1h). All results indicate that miR-181a suppresses the proliferation of SMSCs.

### 3.2. MiR-181a Facilitates SMSCs’ Differentiation

In order to reveal the regulating function of miR-181a on SMSCs’ differentiation, we induced SMSCs to differentiate. We found that the content of miR-181a increased first and then decreased during the differentiation process (Figure 2a,b), which demonstrated that miR-181a may participate in SMSCs’ differentiation. We then detected the expression of MyHC, MyoD, and MyoG after SMSCs were treated with miR-181a mimic, inhibitor, and NC. We found that the mRNA expression of MyHC, MyoD, and MyoG and the protein expression of MyHC in SMSCs treated with miR-181a mimic were higher than the NC group (Figure 2c,e,g). In contrast, transfection of miR-181a inhibitor significantly suppressed the mRNA expression of MyHC, MyoD, and MyoG and the protein expression of MyHC in SMSCs (Figure 2d,f,h). In addition, transfection of miR-181a mimic facilitated myotube formation (Figure 2i), while miR-181a inhibitor suppressed myotube formation (Figure 2j). The results showed that miR-181a facilitated SMSCs’ differentiation, while the knockdown of miR-181a suppressed SMSCs’ differentiation. All results indicate that miR-181a can facilitate the differentiation of SMSCs.

### 3.3. YAP1 Is a Target Gene of miR-181a

To further understand the molecular function of miR-181a in the process of proliferation and differentiation of SMSCs, first, we checked the mature sequence of miR-181a and found it is conserved in different species (Figure 3a). Then, we predicted the potential target gene of miR-181a using RNAhybrid online software, and found that miR-181a could bind to the 3′UTR region of the YAP1 gene (Figure 3b). We also found that the combination of the two is a stable form according to the potential interaction model (Figure 3c). These findings suggest that YAP1 may be a target gene. Therefore, we performed dual-luciferase reporter gene detection in HEK293T cells to confirm whether the YAP1 gene is the target gene (Figure 3d). We mixed the wild-type vector (PMIR-YAP1-3′UTR-WT) or the mutant vector (PMIR-YAP1-3′UTR-MT) with miR-181a mimic or NC, respectively. Then, we transfected the mixture into HEK293T cells. When we mixed the wild-type vector with miR-181a mimic, we found that the luciferase activity was significantly lower than the group with mimic NC. When we mixed the mutant-type vector with miR-181a mimic or mimic NC, the luciferase activity was no different. We also detected the mRNA and protein expression levels of YAP1 in SMSCs. We found they were significantly decreased after overexpression of miR-181a (Figure 3e,g,i). At the same time, we found YAP1 mRNA and protein expression upregulation after inhibition of miR-181a. (Figure 3f,h,j). All these findings indicate that YAP1 is a target gene of miR-181a.

## 4. Discussion

In our study, we explored the function of miR-181a in the proliferation and differentiation of Hu sheep SMSCs. In the RNA-seq of prenatal and postpartum skeletal muscles of sheep, Liu et al. detected 1086 known miRNAs and 40 new miRNAs [4]. This indicates that miRNA participates in the growth and development of sheep skeletal muscle. By predicting BMP2 as a potential target gene of miR-378, Lu et al. found that miR-378 can affect the proliferation of myoblasts in sheep [23]. The antagonism of miR-125b and lnc-SMET has been reported to affect sheep muscle growth by controlling the abundance of IGF2 protein [24]. Meanwhile, miR-181a has been reported to widely participate in muscle growth, cancer progression, and disease occurrence [25,26,27,28]. In our study, we detected the expression of miR-181a during the differentiation process, and found that it increases first and then decreases. This result prompted us to speculate that miR-181a might participate in the differentiation process of skeletal muscle satellite cells. This also suggested that miR-181a may be of great importance in the proliferation and differentiation of SMSCs. To better understand the role of miR-181a in the proliferation of SMSCs, qRT-PCR, Western blot, cck-8, flow cytometry, and EDU were used to reveal the regulated function of miR-181a in SMSCs. Finally, we found that miR-181a can suppress the proliferation of SMSCs. To understand the role of miR-181a in the differentiation of SMSCs, we detected the mRNA and protein expression of MyHC, MyoD, and MyoG. These genes are widely considered to play a key role in muscle satellite cell differentiation [29]. The results showed that miR-181a can facilitate the expression of these genes, and that interference with miR-181a can inhibit the expression of these genes. Furthermore, immunofluorescence experiments showed that miR-181a can facilitate the formation of myotubes, and interference with miR-181a can reduce the formation of myotubes. This suggests that miR-181a can facilitate the differentiation of SMSCs. The effect of miR-181a on the differentiation of SMSCs is also in line with previous studies [30]. The above results prompted us to conclude that miR-181a may participate in the growth and development of Hu sheep skeletal muscle.

As we all know, miRNAs can bind to the target gene and regulate its expression [31]. The identification of target genes is therefore helpful for understanding the function of miRNA. MiR-181a has been reported to affect GC cell apoptosis by targeting SIRT1 [32]. MiR-181a can also regulate platelet activation during storage by targeting RAP1B [33]. We found that YAP1 may be a target gene of miR-181a using RNAhybrid online software. Then, we confirmed that YAP1 is the target gene, according to the results of dual-luciferase reporter assay, qRT-PCR, and Western blot. The YAP1 gene is a downstream gene of the Hippo pathway. Accumulated research shows that YAP1 participates in cell proliferation, apoptosis, and migration [34,35]. Through our research, we found that miR-181a can bind to the 3′UTR region of the YAP1 gene through the dual-luciferase report experiment. In addition, we also found that miR-181a can reduce the expression of YAP1 in both mRNA and protein in Hu sheep SMSCs, while interference of miR-181a has the opposite result. All these results indicate that YAP1 is the target gene of miR-181a.

In conclusion, our research shows that MiR-181a suppresses the proliferation of SMSCs, facilitates the differentiation of SMSCs, and targets YAP1.

## Figures and Tables

**Figure 1 genes-13-00520-f001:**
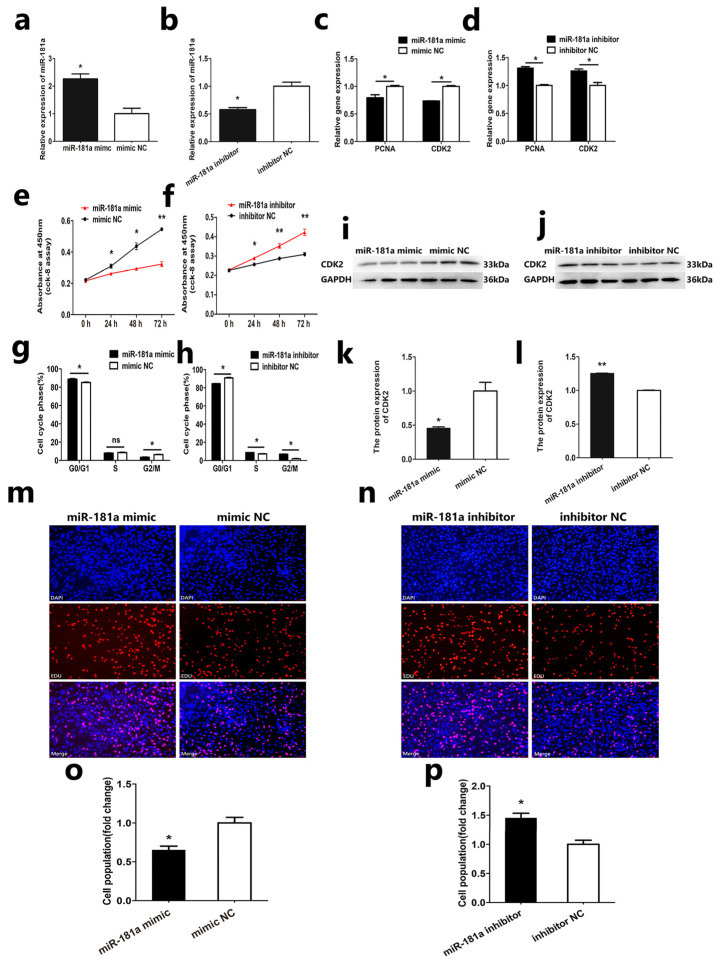
MiR-181a suppresses SMSCs’ proliferation in Hu sheep: (**a**,**b**) The relative expression of miR-181a; (**c**,**d**) The relative mRNA expression of PCNA and CDK2; (**e**,**f**) CCK-8 assay of SMSCs; (**g**,**h**) Cell-cycle analysis of SMSCs; (**i**,**j**) The relative protein expression of CDK2; (**k**,**l**) Statistical analysis of (**i**,**j**); (**m**,**n**) EdU assay of SMSCs. Fluorescence inverted microscope (Nikon, Tokyo, Japan) was used to obtain microscopic images. The magnification is 100 times; (**o**,**p**) Statistical analysis of EDU-stained cells in microscopic images of (**m**,**n**). In all graphs, all data are presented as means ± SEM (standard error of the mean) (*n* = 3). Statistical significance was performed by unpaired Student’s *t*-test. (* *p* < 0.05; ** *p* < 0.01; ^ns^ *p* > 0.05) vs. NC (negative control).

**Figure 2 genes-13-00520-f002:**
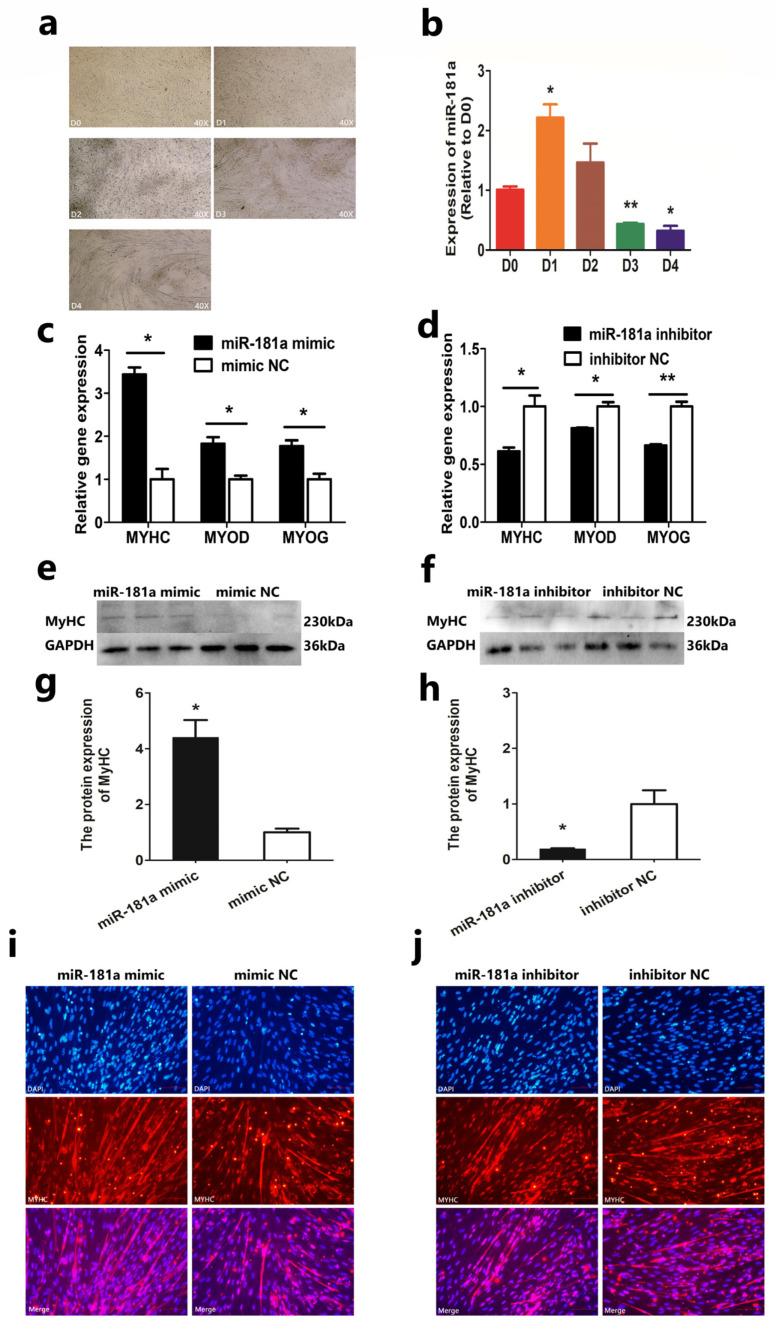
MiR-181a facilitates SMSCs differentiation in Hu sheep: (**a**) Differentiation images of SMSCs in vitro; (**b**) The relative expression of miR-181a during SMSCs’ differentiation (DM0-DM4 represents differentiation from zero to four days); (**c**,**d**) The relative mRNA expression of MyHC, MyoD, and MyoG; (**e**,**f**) The relative protein expression of MyHC; (**g**,**h**) Statistical analysis of (**e**,**f**); (**i**,**j**) Immunofluorescence staining of SMSCs. The magnification is 100 times. In all graphs, all data are presented as means ± SEM (standard error of the mean) (*n* = 3). Statistical significance was performed by unpaired Student’s *t*-test. (* *p* < 0.05; ** *p* < 0.01) vs. NC (negative control).

**Figure 3 genes-13-00520-f003:**
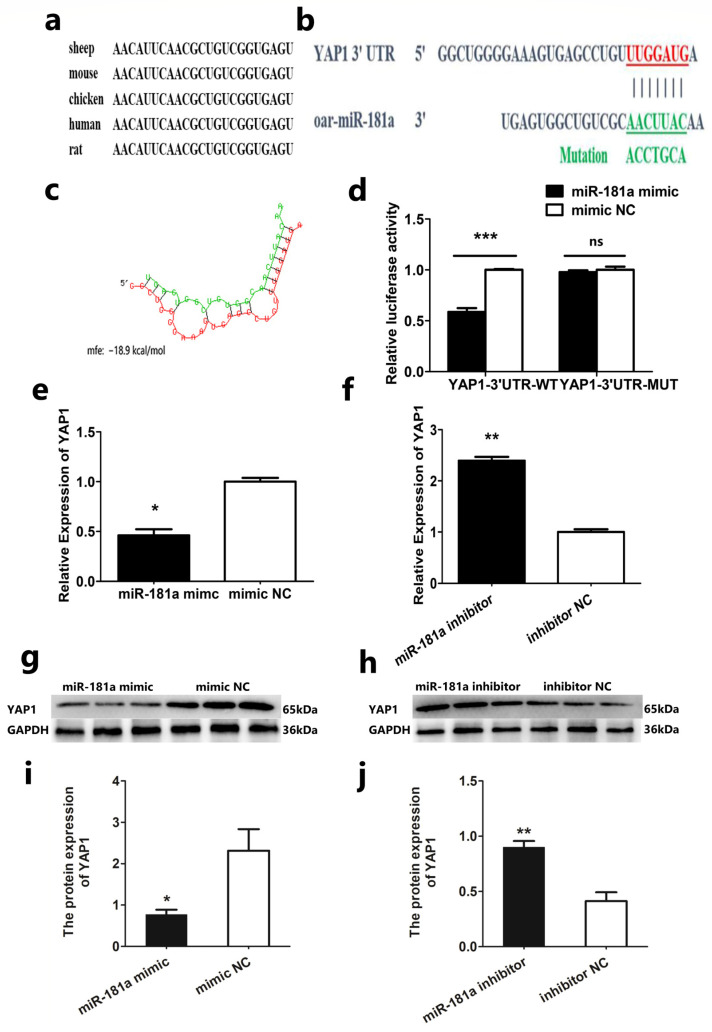
YAP1 is a Target Gene of miR-181a: (**a**) Homology analysis of miR-181a in different species; (**b**) RNAhybrid predicted the targeted binding relationship between YAP1 and miR-181a; (**c**) The interaction model between miR-181a and YAP1 3′UTR by RNAhybrid online software; (**d**) The luciferase assays of HEK293T; (**e**,**f**) The mRNA expression of YAP1; (**g**,**h**) The protein expression of YAP1; (**i**,**j**) Statistical analysis of (**g**,**h**). In all graphs, all data are presented as means ± SEM (standard error of the mean) (*n* = 3). Statistical significance was performed by unpaired Student’s *t*-test. (* *p* < 0.05; ** *p* < 0.01; *** *p* < 0.01; ^ns^ *p* > 0.05) vs. NC (negative control).

**Table 1 genes-13-00520-t001:** MiRNA primers used for qRT-PCR.

Gene	Primer Sequence (5′-3′)	Annealing Temperature (°C)
miR-181a	F: CGAACATTCAACGCTGTCG	58
	R: AGTGCAGGGTCCGAGGTATT	
Stem loop primer	AACATTCAACGCTGTCGGTGAGTGTCGTATCCAG TGCGAATACCTCGGACCCTGCACTGGATACGAC	60
U6	F: CTCGCTTCGGCAGCACA	60
	R: AACGCTTCACGAATTTGCGT	

**Table 2 genes-13-00520-t002:** Gene primers used for qRT-PCR.

Gene	Primer Sequence (5′-3′)	Product Size (bp)	Annealing Temperature (°C)	Accession Number
YAP1	F: GGACTAGTCCAACTATGACGACCAATAGCTCA R: CGACGCGTCGAAATAGTGGATGAAAGAA	108	60	NM_001267881.2
CDK2	F: AGAAGTGGCTGCATCACAAG	92	60	NM_001142509.1
	R: TCTCAGAATCTCCAGGGAATAG			
PCNA	F: CGAGGGCTTCGACACTTAC R: GTCTTCATTGCCAGCACATT	97	60	XM_004014340.4
MYHC	F: TCGTCAAGGCCACAATTTG	101	60	XM_004010325.3
MYOG	R: CTGCTGCAACACCTGGTCCT F: AATGAAGCCTTCGAGGCCC R: CGCTCTATGTACTGGATGGCG	101	60	NM_001174109.1
MYOD	F: GCTCCAGAACCGCAGTAAGTT	106	60	NM_001009390.1
	R: CGGCGACAGCAGCTCCATA			
GAPDH	F: TCTCAAGGGCATTCTAGGCTAC	151	60	NM_001190390.1
	R: GCCGAATTCATTGTCGTACCAG			

**Table 3 genes-13-00520-t003:** Sequences of oligonucleotide.

Fragment Name	Sequence (5′-3′)
miR-181a mimic	AACAUUCAACGCUGUCGGUGAGU
	ACUCACCGACAGCGUUGAAUGUU
miR-181a inhibitor	ACUCACCGACAGCGUUGAAUGUU

**Table 4 genes-13-00520-t004:** Primers of construction of related plasmids.

Primer Name	Primer Sequence (5′-3′)	Product Size (bp)	Annealing Temperature (°C)
YAP1-3′UTR-WT	F: AAAAGATCCTTTATTAAGCTTCTCTTCTTGTCCATTGCCGC	302	60
	R: CATAGGCCGGCATAGACGCGTTAGACCAGTAAGTCATGTTTTCCCA		
YAP1-3′UTR-MT	F: CCTGTACCTGCAATGGATGCCATTCCTTTTGCC	6746	63
	R: TCCATTGCAGGTACAGGCTCACTTTCCCCAGC

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
