# Peer review of "MicroRNA-181a Regulates the Proliferation and Differentiation of Hu Sheep Skeletal Muscle Satellite Cells and Targets the YAP1 Gene"

_genes, 2022, doi:10.3390/genes13030520_

Round 1
Reviewer 1 Report
I read this manuscript with a great interest.
The manuscript is well structured. The introduction shows the importance of mRNA in the post-translational regulation of the genes, summarises the results of MiR181 studies obtained previously. The methods are clearly stated. The research results are described in details, and the conclusions are based on the results obtained.
I have only a few small comments.
Please indicate in the abstract (L 20) that the SMSCs were derived from longissimus dorsi muscle of 3-old months fetus.
I suppose, that word "respectively" at the end of the sentence (L57-58) is not necessary.
I believe that the manuscript will be very interesting for the readers of Genes.
Reviewer 2 Report
Dear authors,
This study analyses the microRNA 181a function in development of Hu sheep skeletal muscle. The manuscript is well written and structured, the introduction provides sufficient background, the results are clearly presented, and the conclusions are supported by the results. However, major changes are necessary before its publication:
Line 101: change “penicil-lin” to “penicillin”.
Line 103: change “extractio” to “extraction”.
Line 118: If the GAPDH was used as a housekeeping, What do the authors use the U6 gene for? Since the reference genes vary their expression depending on the species and tissue analyzed, the authors should explain (and add bibliographical references) that demonstrate that the GADPH gene is stable in this species and tissue analyzed.
Line 223: Before carrying out a parametric test, such as a t-student, the authors must ensure that their data follow a normal distribution and show homoscedasticity. If this were not the case, a parametric test could not be applied for the analysis.
Line 224: Change “p<0.05 or p<0.01” to “p<0.05”. The p-value must be established before performing the statistical analysis and must be unique. Therefore, the authors must choose whether the p-value for their analysis is 0.05 or 0.01, not both.
Figure 1: The authors should improve the quality of the figure.
Round 2
Reviewer 2 Report
The authors have made all the requested changes.
This manuscript is a resubmission of an earlier submission. The following is a list of the peer review reports and author responses from that submission.
Round 1
Reviewer 1 Report
In the paper "MicroRNA-181a regulates the proliferation and differentiation of Hu sheep Skeletal muscle satellite cells and targets the YAP1 gene" He et al describe the expression and molecular mechanism of action of miR181a.
The authors focus on the molecular circuit through which miR181a and YAP1 regulate SMSCs differentiation, however they should clarify the following major points.
Western Blots in Fig 1e-f show PCNA protein levels, while in the text CDK2 protein levels are described. Moreover, a protein quantification is needed in order to understand if the changes in protein level are statistically significant.
Graphs in Fig 1 i-j don't have a clear y-axis label. Do they refer to the quantifications of the BrdU experiments shown in Fig. 1g-h? Maybe the authors could represent the data as "% of BrdU+ cells".
Lines 229-230: the stating "which indicated that miR-181a was involved in SMSCs differentiation" is too strong. It is better to say that "miR181a could be involved in SMSCs differentiation", at this point.
Statistical analysis is missing in Fig.2a
Fig. 2d-e: a protein quantification is needed in order to understand if the changes in protein level are statistically significant.
The authors write that the miR181a promotes SMSCs differentiation, but they only analyze the mRNA and protein levels of a few differentiation markers. Either the authors change the conclusion of this paragraph to "miR181a could promoting SMSCs differentiation", or they add functional tests to verify the role of miR181a on SMSC2 differentiation in vitro.
Fig. 3e-f: beyond the YAP1 mRNA analysis by qRT-PCR, the authors should add the western blot and relative protein quantification of YAP1 after miR181a knock-down and overexpression, to actually see significant changes in YAP1 protein, which can be translated into changes in YAP1-dependent molecular circuits.
Paragraph 3: the only luciferase assay using the exogenous luciferase-expressing vectors and the transfection of miRNA mimics and inhibitors is not enough to establish whether a regulatory circuits exists between YAP1 mRNA and miR181a in SMSCs. The authors should at least verify whether miR181a binds YAP1 mRNA in SMSCs by an RNA-pulldown assay.
Moreover, how is the trend of YAP1 mRNA and protein during SMSCs differentiation? It should have an opposite trend than miR181a, if a regulatory circuits really exists between these two molecules and if it has a functional role on SMSCs differentiation.
Reviewer 2 Report
In the paper by He et al., the authors explore the role of miR-181a in the proliferation and myogenic differentiation of skeletal muscle satellite cells (SMSCs) derived from a Chinese sheep breed. They showed that miR-181a inhibited the proliferation and promoted the differentiation of SMSCs. Furthermore, YAP1 was confirmed as a target gene of miR-181a. This work could be a valuable contribution, not only to the Special Issue "Molecular Mechanism Analysis of Important Traits and Breeding of New Breeds of Sheep", but also from a general point of view to gain insight in the role of post-transcriptional regulation mediated by miRNAs in skeletal muscle development. However, even if the findings are potentially interesting, the work is still preliminary/incomplete for publication.
Specific points that could be addressed:
1) The introduction should be improved to make the argument construction more understandable. Besides, considering that the study is performed in sheep, papers published in other ruminant species on the regulation of the development of skeletal muscle by miRNAs may be of interest to the readership and, therefore, they should be cited somewhere (e.g. https://doi.org/10.1007/s11626-016-0109-y; https://doi.org/10.1038/s41598-018-22262-4).
2) Please, revise the English language throughout manuscript. Some examples:
- The use of “meanwhile” and “In addition” in lines 35 and 42, respectively.
- Sentences like “More and more studies have shown that 55 MiR-181a plays an important role in many life activities” (Line 55) or “As we all know, miRNAs usually regulate mRNA expression by […] (Line 294).
- Correct: “The SMSCs were inoculated into cell culture dishes added with a growth medium” (Lineas 152, 159, 168). The cells were seeded (not inoculated) in DMEM-F12 medium (not added with medium).
3) The main cell culture model used in this work are SMSCs derived from a 3-month-old Hu sheep fetus. Authors should provide evidence of their characterization of the established SMSCs (gene expression profiling including Pax7) and they must confirm the cells’ differentiation capacity after myogenic induction conditions (read also comment 8).
4) Authors state that they “test the effects of overexpression and interference of miR-181a on […]”. Therefore, miR-181a expression levels after the transfection of mimics, inhibitors, negative control/scramble, and in non-transfected cells, must be shown at different time points after transfection. Please, consider to represent the expression levels -instead of the mRNA fold increase- for figures 1a,b, in order to facilitate the reader the comparison between mimic and inhibitor.
5) Western blots in Figure 1e,f must be quantified and normalized to GAPDH loading control. Please, note that there is a mistake either in the text or in labelling the blots, since the text refers to CDK2 protein expression and the figure and figure legend to PCNA protein. Amend the protein molecular weight if necessary.
6) It is not clear from the text if figures 1i and 1j correspond to image quantification from 1g and 1h, respectively. Assuming that is the case, it seems difficult to infer the numerical difference represented in the graph when looking at the pictures, unless they are not the most representative ones. This must be clarified. Instead of indicating the number of regions counted, it may be more informative to know the number of cells counted (“at least N cells from each independent experiment were counted for each group) and to represent the % of EdU+ cells. In addition, authors should label the pictures in Figure 1c (DAPI in blue, EdU in red) and include the information in the figure legend. The methods regarding these experiments should also be revised since the incubation time to allow for EdU incorporation into the DNA of cells undergoing DNA synthesis is not mentioned.
Considering data in 1g,i, and that EdU is used to mark cells traversing S-phase, how do the authors explain the results in figure 1k? (it would be nice to see in the figure the cell cycle profiles after Modfit analysis). Have the authors tried to evaluate the effect on cell proliferation at different time points? In the methods, it is stated that 60% confluent cultures were transfected and that the analysis were performed 48 h later. Based on working experience, it would be reasonable to think that the proliferation was pretty reduced at this time point due to the high confluency.
7) The myogenic differentiation protocol is poorly described. The paper would have clearly benefited from presenting data on morphological features and marker gene expression patterns along differentiation of non-transfected controls and the transfected cells (mimics, inhibitors, scrambled). This not only would corroborate the myogenic properties of the established SMSCs (see comment 3) but would also support the results. For example, how do the authors explain the extremely low protein expression of the terminal differentiation marker MyHC under scrambled conditions? Immunofluorescence analysis could help to interpret the results.
With regard to the differentiation experiments, there is no description in the manuscript of the time points at which mRNA and protein expression in Figure 2b-e was evaluated. Clarification of this point is important, since the kinetics of miR-181a expression (Figure 2a) could suggest that it is involved in the early steps of differentiation, but not in the maintenance of the mature phenotype. The reported results should be discussed in comparison to other reports describing miR-181 involvement in establishing the muscle phenotype (https://doi.org/10.1038/ncb1373).
8) For clarity reasons, marker expression could be displayed and discussed following the myogenic differentiation pathway: activation of quiescent satellite cells, myogenic commitment and fusion to form mature myotubes. It would be interesting to see p21 expression in these experiments. Again, western blots in Figure 2d,e must be quantified and normalized to GAPDH loading control.
9) One of the major interests of this study is in describing YAP as novel target for miR-181. Unfortunately, this aspect is not further explored in the paper and experiments establishing a link between Hippo signaling pathway and the other reported results are missing. In addition, to validate these results and prompt the interest of a broad readership, the authors need to perform the experiments in another cell model (e.g. the widely used myoblast cell line C2C12), since Figure 3 experiments are only performed in HEK293T cells.
10) Finally, to sustain the relevance of the identified signalling in the context of the molecular breeding of the Hu sheep, a connection must be established between the series of experiments conducted in vitro and the situation in vivo. At least, an evaluation of miRNA-181a expression in sheep muscle must be performed: analysis in different muscle tissues, different time points, etc. (as a reference, see the experimental samples used by Liu et al. in the RNA-seq study where they showed differential expression of miR-181a).
Round 2
Reviewer 2 Report
The authors have not convincingly addressed some of the raised concerns and they have only responded to a certain extent to both reviewer’s comments.
Author Response
Thank you very much for all the suggestions for this article, which have improved the manuscript. We have further revised the manuscript. The introduction and conclusion have been further modified. Thank you again for your work!
